# Antimicrobial Peptides and Cationic Nanoparticles: A Broad-Spectrum Weapon to Fight Multi-Drug Resistance Not Only in Bacteria

**DOI:** 10.3390/ijms23116108

**Published:** 2022-05-29

**Authors:** Giulia E. Valenti, Silvana Alfei, Debora Caviglia, Cinzia Domenicotti, Barbara Marengo

**Affiliations:** 1Department of Experimental Medicine (DIMES), General Pathology Section, University of Genoa, 16132 Genoa, Italy; giuliaelda.valenti@edu.unige.it (G.E.V.); barbara.marengo@unige.it (B.M.); 2Department of Pharmacy, University of Genoa, 16148 Genoa, Italy; alfei@dictfa.unige.it; 3Department of Surgical Sciences and Integrated Diagnostics (DISC), University of Genoa, Viale Benedetto XV, 6, 16132 Genova, Italy; debora.cavigliad@edu.unige.it; 4Inter-University Center for the Promotion of the 3Rs Principles in Teaching & Research (Centro 3R), 56122 Pisa, Italy

**Keywords:** multi-drug resistance, antibiotics, anticancer drugs, antimicrobial peptides, cationic nanoparticles

## Abstract

In the last few years, antibiotic resistance and, analogously, anticancer drug resistance have increased considerably, becoming one of the main public health problems. For this reason, it is crucial to find therapeutic strategies able to counteract the onset of multi-drug resistance (MDR). In this review, a critical overview of the innovative tools available today to fight MDR is reported. In this direction, the use of membrane-disruptive peptides/peptidomimetics (MDPs), such as antimicrobial peptides (AMPs), has received particular attention, due to their high selectivity and to their limited side effects. Moreover, similarities between bacteria and cancer cells are herein reported and the hypothesis of the possible use of AMPs also in anticancer therapies is discussed. However, it is important to take into account the limitations that could negatively impact clinical application and, in particular, the need for an efficient delivery system. In this regard, the use of nanoparticles (NPs) is proposed as a potential strategy to improve therapy; moreover, among polymeric NPs, cationic ones are emerging as promising tools able to fight the onset of MDR both in bacteria and in cancer cells.

## 1. Bacterial Multi-Drug Resistance: A General Overview

The discovery and use of antibiotics, starting from the second half of the twentieth century, has revolutionized modern medicine, thus allowing the treatment and prevention of infectious diseases and drastically improving people’s quality of life.

However, in the last few years, the phenomenon of antibiotic resistance has increased considerably, becoming nowadays one of the main public health problems worldwide. This entails important clinical implications (increase in morbidity, lethality, duration of the disease, possible development of complications and epidemics) and additional costs too, required for the use of expensive drugs and procedures, for the lengthening of hospital stays and for any disability. In fact, the continuous use of antibiotics promotes the onset and the diffusion of multi-drug resistance (MDR), further reducing the possibility of an effective treatment and increasing mortality [1]. In addition, while, until a few years ago, the presence of bacterial MDR was limited to the hospital environment, nowadays, it is widespread and involves several species of Gram-positive bacteria, such as *Staphylococcus aureus*, *Enterococcus faecium*, *Enterococcus faecalis* and *Streptococcus pneumoniae* and of Gram-negative pathogens, including the family of *Enterobacteriaceae*, and non-fermenting *Acinetobacter baumannii*, *Klebsiella pneumoniae* and *Pseudomonas aeruginosa* [2].

Resistance may be due to intrinsic or acquired mechanisms [3]. While the former occurs naturally, the latter are a consequence of the stress induced by excessive treatment with antibiotics and may be due to genetic or epigenetic alterations [4,5]. In fact, antibiotic resistance may be the result of spontaneous mutations with the acquisition of resistant genes [6,7,8] and of several biochemical alterations, such as the decrease in drug efflux pump activity and membrane porins, leading to a reduction in the intracellular drug concentration [9,10] and drug inactivation through enzymatic action [11,12] or the modulation of metabolic pathways [13,14,15].

Therefore, since antibiotic resistance is one of the major public health problems today, it is crucial to find new strategies able to counteract it.

## 2. Sometimes They Come Back: Old Strategy to Defeat the Resistance

Although many novel molecules with antimicrobial activity have been synthesized, only a few are undergoing clinical trials and are active against MDR pathogens [16]. For this reason, research has focused on finding other alternative strategies. Various approaches have been proposed, such as the use of “old” antibiotics to which MDR bacteria should no longer be resistant [17], or even more specific ones obtained by changing the molecular structure to restore their activity, as in the case of resistance to vancomycin [18]. 

Antibiotic adjuvants have been developed and used in therapy to restore the activity of existing drugs no longer functioning, such as, for example, the β-lactamase inhibitors, including clavulanic acid, β-lactam-like sulfones (sulbactam and tazobactam), diazabicyclooctanes (avibactam and relebactam) and boronic acids (vaborbactam) [19]. In fact, unfortunately, the appearance of new variants of β-lactamase enzymes and the emergence and spread of the ultra-broad-spectrum metallo-β-lactamases of class B dramatically reduce the effectiveness of this strategy [19]. Other approaches to overcome MDR include the use of inhibitors of drug efflux pumps and outer membrane permeabilizers able to promote antibiotic uptake and to improve their concentration at target sites [20,21], as well as the employment of membrane disruptor molecules. In this context, the compound 2-((3-[3,6-dichloro-9H-carbazol-9-yl]-2-hydroxypropyl) amino)-2-(hydroxy-methyl) propane-1,3-diol (DCAP) is an antibacterial agent that causes cell lysis by altering membrane potential and permeability. It acts specifically on the membranes of bacteria and has a broad spectrum of action on both Gram-positive and Gram-negative pathogens [22].

Recently, the efficacy of Odilorhabdins against MDR bacteria has been proposed. These compounds interfere with protein synthesis by binding to the small subunit of bacterial ribosomes, which is different from the target subunit of the traditional and no longer active antibiotics [23]. Moreover, it has also been reported that the benzoimidazole moiety, already used for the treatment of other diseases, inhibits peptide deformylase and thus bacteria protein synthesis [24,25] and could represent a novel pharmacophore for the synthesis of new antibiotic agents.

The use of metal-based antibacterial agents has also given promising results, exploiting the multiple mechanisms of action of metal complexes, mainly with ruthenium, gallium, bismuth, silver and copper [26,27]. Another strategy involves the use of bacteriophages, viral agents that exclusively infect bacteria, exerting bactericidal activity; these could have a synergistic action when given in combination with antibiotics [28,29].

## 3. New Approaches to Defeat the Resistance

The high incidence of the selection of antibiotic-resistant bacterial strains highlights the urgent need to develop effective therapeutic strategies aimed to counteract this phenomenon. In this context, the use of membrane-disruptive peptides/peptidomimetics (MDPs) is receiving particular attention and, among these, antimicrobial peptides (AMPs), both of natural (NAMPs) and synthetic (SAMPs) origin, seem to be promising [30,31]. NAMPs are short cationic peptides that play a fundamental role in the innate immunity [32,33,34]. The amphiphilic structure and the presence of a net positive charge are the most important characteristics that typify NAMPs and that influence their mode of action [35,36]. In particular, NAMPs, as with other cationic materials, act mainly with a rapid and non-specific disruptive action on bacterial membranes, not allowing the pathogens to organize adaptive processes for becoming resistant [37]. Therefore, it is clear that NAMPs, in order to exert their antibacterial action, must be able to easily and firmly interact with the bacterial membranes. In this regard, it has been widely recognized that the electrostatic interactions between NAMPs and the negatively charged lipids of the outer envelope of Gram-negative bacteria and of the surface of Gram-positive bacteria are the main factors responsible for the higher selectivity of NAMPs towards bacteria [38,39,40,41,42,43,44,45,46,47]. Notably, the membrane of eukaryotic cells is mainly composed of lipids with zero net charge, thus drastically lowering the probabilities of electrostatic interaction with cationic materials (Figure 1) [38,39,40,41,42,43,44,45,46,47]. Additionally, mammalian cells are less susceptible to the detrimental effects of NAMPs due to the presence of high amounts of cholesterol, which contributes to stabilizing the membrane (Figure 1) [48].

However, although the primary target of NAMPs is the lipid matrix of the bacterial cell membrane [46], it has been reported that, thanks to their amphiphilic characteristic, several NAMPs also enter the bacterial cells and exert their toxic effect by impairing DNA, RNA and enzymes, thus enhancing their original therapeutic efficacy [49,50,51].

The reduced ability of NAMPs to induce resistance and to exert toxic effects on healthy cells has encouraged the development of SAMPs, which, compared to NAMPs, possess a more easily modifiable chemical structure, allowing one to obtain molecules capable of interacting more strongly with the surface of bacteria, thus achieving greater antimicrobial activity, better physicochemical stability, and greater compatibility with drug delivery methodologies [52]. As for NAMPs, for SAMPs, the determining factor responsible for their antibacterial action is the amphiphilicity and cationic character [53,54,55]. Moreover, it has been found that the enhanced interactions of SAMPs with the negatively charged lipids of bacterial membranes not only induce membrane damage, but also favor the entry of synthetic peptidomimetics into bacteria [56,57], thus exerting a toxic action at the intracellular level [58].

However, the clinical application of these SAMPs as new antibiotics is still hampered by high production costs, elevated hemolytic toxicity and a reduced half-life in serum [59,60,61,62,63,64]. 

## 4. Multi-Drug Resistance: A Common Feature of Bacteria and Cancer Cells

### 4.1. Bacteria and Cancer Cells: What Do They Share?

A very similar phenomenon of drug resistance has been observed in the treatment of various kinds of cancer. In fact, after an initial positive response of the anticancer therapy, the onset of MDR and the negative prognosis occurs [65,66]. Cancer cells have been found to acquire an MDR phenotype through several mechanisms, including increased expression of MDR transporters, which increase drug efflux and/or reduce its uptake; defects in the apoptotic program; promotion of the autophagy process; alteration of drug metabolism and cancer cell metabolism and alteration of intracellular redox homeostasis [67,68,69,70,71,72,73,74]. Notably, many of these mechanisms are the same as those that allow bacteria to develop antibiotic resistance. However, cancer cells, similarly to bacteria, have negatively charged lipids in their cell membranes [75], while healthy mammalian cells have neutrally charged lipids in the cell membrane (Figure 2) [76].

Moreover, using NAMPs as template molecules, SAMPs are continuously developed to produce de novo scaffolds that resemble the parent peptides, which can maintain therapeutic effectiveness with higher biological stability and a greater safety profile. Therefore, it is possible to hypothesize that NAMPs, as well as SAMPs, by selectively interacting with the negatively charged lipids of the tumor cell membrane and preserving healthy cells from injury, might be a promising approach to counteract cancer chemoresistance [77]. 

### 4.2. Anticancer Peptides to Fight Cancer Chemoresistance

Several structural and functional studies led to the evidence of cationic low-molecular-weight peptides having antitumor activity, which were then classified as anticancer peptides (ACPs) [78,79,80,81,82]. In fact, cationic peptides, isolated from various organisms and historically known for their antimicrobial activities, were studied and described as potent anticancer agents in 1985 [83]. The use of these peptides to treat cancer has received considerable support since, as mentioned, cancer cells have negatively charged membrane lipids, which are the elective targets of these compounds (Figure 2) [84,85,86]. In addition, cancer cell membranes have many microvilli and therefore a greater surface for interaction with ACPs in comparison with healthy cells (Figure 2) [77,87]. 

Another factor that could influence the cancer cell response to ACPs is the membrane cholesterol content. In fact, it has been reported that cancer cells (breast and prostate cancers) having high cholesterol levels are protected from the action of APCs, while other ones (leukemia and lung cancer) with low cholesterol content are more susceptible to the ACP-mediated lytic action [88,89,90]. 

Additionally, ACPs can induce cancer cell death by disrupting mitochondrial membrane integrity, leading to cytochrome C release and apoptosis induction [91,92,93,94,95,96,97,98]. Moreover, it has been demonstrated that ACPs can exert their anticancer activity by activating other mechanisms, such as immunogenic cell death [99], inhibition of DNA polymerase [100] and an anti-angiogenic action [101]. 

However, although the potential for the clinical use of ACPs is promising, it is necessary to take into account that all the above-mentioned characteristics of cancer cells can interfere with the therapeutic efficacy of this approach.

## 5. Clinical Development of AMPs and ACPs: An Open Issue

Based on the consideration that both AMPs and ACPs are not traditional drugs and that bacterial and cancer cells share some structural features, the clinical development of both AMPs and ACPs faces similar challenges and limitations. For this reason, we have focused our attention on the factors that could limit the clinical development of AMPs, reasonably assuming that such issues could, potentially, also be found using ACPs.

Firstly, since NAMPs have structures that are not optimized for a specific antibacterial or anticancer action, a fundamental step for their clinical development is their structural optimization. Other critical issues that can limit the employment of NAMPs consist in their poor pharmacokinetic (PK) properties, the high cost of synthesis, their inability to recognize specific receptors and the reduced correlation between in vitro results and those found in animal models [37,102,103,104,105,106]. 

However, parenterally administrable SAMPs with optimized structures have been already successfully developed [104,105,107,108]. Moreover, although the peptidic nature of AMPs precludes their oral delivery, various specific delivery systems have been engineered to escape the degradative attack of digestive enzymes and to increase their intestinal absorption [77]. Several strategies can be used to enhance the PK properties of AMPs, including the use of the D-enantiomers constituting amino acids to increase stability [109], and the packaging of AMPs in inactive liposomes targeting a specific cancer cell ligand in order to specifically deliver the active peptide [110,111,112,113,114]. 

With regard to the high cost of production, the problem may be overcome by the application of modern production technologies, such as solid-phase synthesis, and by synthesis on a large scale. Furthermore, the additional difficulty related to the fact that AMPs select their targets by non-receptor-mediated recognition and rather through the membrane properties may represent a strength point since it leads to a lower likelihood of developing resistance. 

Despite the above-reported problems, several ACPs show antitumor effects (Table 1) due to their unique mechanism of action, to their relatively high tissue penetration and to the low acquisition of drug resistance; they have many advantages over conventional chemotherapy and some of them are currently being studied for clinical application in oncologic patients (Table 2) [115]. 

### 5.1. Insight into the Antitumor Activity of ACPs

#### 5.1.1. ACPs as Membrane Disruptors

It is universally recognized that ACPs act mainly as membrane-active molecules and two models of action have been described. The first one is the “barrel-stave” model [138], in which ACPs induce pore formation in the cell membrane, by rendering cancer cells unable to maintain normal osmotic pressure (Figure 3). Subsequently, the “carpet” model has been proposed, according to which ACPs induce cell death by destroying the cell membrane, thus causing the massive leakage of cytoplasmic content [139], as observed in melanoma, gastric cancer, liver cancer and cervical cancer cells treated with HPRP-A1-TAT, a hybrid peptide [140] (Figure 3). Another example is Temporin-La, which exerts its anticancer activity by destroying the cell membrane [141]. However, the detailed mechanism of interaction between ACPs and the cell membrane needs to be further studied.

#### 5.1.2. ACPs: Other Mechanisms of Action

More in-depth studies have demonstrated that ACPs can also lead to the apoptosis of cancer cells by destroying the mitochondrial membrane and thus favoring the release of cytochrome C. Such a mechanism has been observed, for example, for Ra-V, Dolastatin 10 (*Dolabella auricularia*) and the *Bacillus subtilis* lipopeptide. Other peptides, such as KV11, FN070315 and Temporin-1CEa, exert their anticancer activity by inhibiting tumor angiogenesis [142,143,144]. Moreover, immune regulation is the main mechanism of action of Bovine lactoferrin (LfcinB) and MENK (an endogenous neuropeptide) [145,146]. Other studies have shown that MENK acts as an immune booster and can inhibit the proliferation of human cancer cells through cyclin-dependent kinase inhibition pathways [147,148,149]. 

Collectively, although the development of adequate systems able to facilitate their delivery remains one of the limiting problems, to date, AMPs potentially possess all the essential features to become the only class of drugs able to fight MDR both in bacterial infections and in cancer. 

## 6. Nanoparticle (NP)-Based Delivery Systems: A Solution for the Clinical Application of AMPs?

To make possible the clinical application of AMPs and ACPs, the development of appropriate delivery systems is fundamental and several strategies to design favorable formulations of these peptides have been developed. Among these, nanoscale delivery systems (liposomes, micelles, and NPs) appear to be valid candidates due to their high biocompatibility. Moreover, such systems are also able to improve drug bioavailability by reducing its early elimination and permitting its gradual and sustained release [76]. Concerning this, it is possible to manufacture stimuli-responsive nanocarriers, whose drug release is controlled and triggered by changes in pH or temperature, by alterations in redox state or by the presence of specific microenvironment components. In addition, several NPs offer the possibility of modifying their surface by using poly-(ethylene glycol) (PEG) to increase the NP blood circulation half-life [150,151]. These include the materials used to prepare such NPs, including natural lipids, proteins, and carbohydrates, as well as synthetic polymers, dendrimers, metals, etc. [152,153]. Unlike free drugs, those formulated in NPs have reduced toxicity toward healthy tissue, can be delivered to specific targets, possess improved pharmacological properties, and allow simultaneous combined administration. 

### 6.1. Use of NP-Based Delivery Systems to Overcome Antibiotic Resistance

To improve the therapeutic efficacy of antibiotics and prevent the acquisition of resistance by bacteria, the use of NPs has been proposed as a promising strategy [151,154,155]. To this end, several NPs able to specifically direct antibiotics to bacteria and to prolong their half-life have been developed. Recently, anionic, functionally mesoporous silica NPs able to encapsulate high amounts of polymyxin B, a long-established NAMP, maintaining its antibacterial activity and improving its biocompatibility, have been formulated [156].

Moreover, azithromycin (AZM)-conjugated clustered NPs (AZM-DA NPs), able to release secondary AZM-conjugated PAMAM NPs (PAMAM-AZM NPs) upon disassembling under an acidic biofilm microenvironment, have been reported [157]. In particular, the positive charge and the small size of these PAMAM-AZM NPs were fundamental to improve their penetration and retention within the biofilm produced by bacteria, to make the bacterial cell more easily permeable and to allow the increased uptake of AZM.

### 6.2. Use of NPs to Counteract Cancer Chemoresistance

Although the research on anticancer polymer/peptide nanocomposites is a recent field of interest, the results obtained so far seem promising. In fact, anticancer polymer/peptide NPs have been shown to be able to reduce the onset of chemoresistance, to inhibit tumor growth and to limit the formation of metastases in vivo [31]. In addition, it has been observed that such anticancer polymer/peptide NPs not only exhibit improved retention times and permeation effects in tumor tissues, but also reduced toxicity towards healthy tissues in respect to non-polymer-modified peptides [31]. Specifically, it has been reported that several polymer-based NAMP- or SAMP-loaded NPs, by facilitating cell uptake, provided enhanced therapeutic effects. Notably, a pH-sensitive ACP-loaded polymeric micelle has been developed by conjugating the synthetic poly (amino ester)-poly-(ethylene glycol) copolymer to the therapeutic peptide CGKRKD (KLAKLAK), to increase the intracellular ACP concentration [158]. Furthermore, in the study of Li’s group, melittin (MLT)-loaded zeolitic imidazolate framework-8 (MLT@ZIF-8) NPs able to improve MLT stability and to inhibit its hemolytic activity were developed [159]. In addition, in respect to free MLT, this nano-formulation displayed enhanced antitumor effects, due to the increased uptake of the drug [159].

### 6.3. Cationic NPs: Not Only Delivery Systems

Among polymeric NPs, cationic ones are emerging as macromolecules that could play an important role against MDR bacteria, as well as MDR cancer cells. As previously reported, one of the causes of antibiotic resistance is the reduced or nullified capability of the drugs to penetrate bacterial membranes [160]. Gram-positive bacteria—and, even more so, Gram-negative ones—have highly organized external membranes with negative charges [161]. In this context, cationic NPs, used to encapsulate AMPs, are capable of selectively interacting with the net anionic charges of bacterial membranes, rather than with the zwitterionic ones of eukaryotic cells, and considerably reduce AMP-mediated toxic effects [162]. Additionally, cationic NPs can form complexes with other macromolecules with negative charges, such as DNA or proteins exerting intrinsic antibacterial and anticancer activity [163]. A great advantage of cationic NPs, with respect to cationic lipids, is represented by the smaller size and the higher loading capacity, thus facilitating their internalization by target cells. Furthermore, being highly cationic, they are completely water-soluble [164].

A very large number of positively charged NPs and nanogels are currently used for various purposes, including the encapsulation of AMPs, and there are many production procedures and/or polymerization methods, which mainly involve either the use of cationic monomers to prepare polymeric NPs or nanogels or cationic reagents to be covalently bound or absorbed to previously prepared NPs or nanogels [165]. 

The most efficient and used techniques to synthesize and produce polymeric NPs are those related to hetero-phase polymerization processes, including emulsion polymerization [166]. To obtain cationic NPs, a combination of different reagents is necessary, such as cationic initiators, monomers, polymers, and surfactants [165]. The choice of monomers or polymers to use depends on the type of the desired charge, which could be permanent or pH-dependent. The most used monomers are vinyl-pyridines and quaternary ammonium cationic monomers, while polymers include, but are not limited to, ethyl methacrlates. Notably, even if the synthesis of cationic NPs is mainly surfactant-free, in some cases, surfactants are used since they help to solubilize the monomers by forming micelles and stabilize the polymer during formation. Additionally, surfactants have also a fundamental role in the size and stability of the obtainable NPs and, therefore, the amount of these compounds is crucial for successful preparations [167]. The most common cationic surfactants used are quaternary ammonium salts such as hexadecyltrimethylammonium- or dodecyltrimethylammonium-bromide, as well as cetyltrimethylammonium chloride [168]. 

#### 6.3.1. Antibacterial Function of Cationic NPs

Several innovative strategies involving cationic NPs have been developed to counteract MDR bacteria. Notably, Hou et al. have selected from chitosan-grafted oligolysine chains cationic NPs that spontaneously self-assemble in water, with great efficacy both in vitro and in vivo against some methicillin-resistant *Staphylococcus aureus* and MDR *E. coli* and *P. aeruginosa* [169]. These NPs, with multiple positive charges in their polymer chains and a high charge density, are able to form hydrogen bonds between chitosan chains and bacteria, leading to membrane cell injury and thereby exerting their bactericidal activity (Figure 4).

The same mechanism of membrane disruption has been reported for branched polyethyleneimine-functionalized silver nanoclusters, which exert a bactericidal effect against MDR infections, thanks to the synergy of silver’s antimicrobial activity with cationic NPs’ selectivity towards bacterial membranes, resulting in low toxicity for mammalian cell membranes [170]. In this context, Chen et al. have developed NPs of metal–organic frameworks incorporated with titanium, which are able to increase reactive oxygen species (ROS) production and are employed for the photodynamic therapy of chronic wounds infected by MDR bacteria [171]. 

Two years ago, three non-cytotoxic dendrimers of fifth generation (G5D) were synthetized and proven to be effective against non-fermenting MDR Gram-negative species such as *P. aeruginosa*, *S. maltophilia* and *A. baumannii* [172]. More recently, a lysine-modified cationic polyester-based dendrimer (G5-PDK), capable of electrostatically interacting with bacterial surfaces, thus leading to membrane injury, has been synthetized and characterized [173]. G5-PDK was demonstrated to have bactericidal effects specifically against isolates of the *Acinetobacter* genus, including *A. baumannii*, responsible for difficult-to-treat nosocomial infections [173], but also against different species of the *Pseudomonas* genus [174]. In this latter study, it was established that the antibacterial potency of G5-PDK depended on the capability of *Pseudomonas* to produce cationic or anionic pigments, thus confirming the above-mentioned mechanism of action of cationic NPs and their high selectivity for bacteria [174]. 

As previously reported, the overall merit of the cationic NPs consists of their intrinsic antibacterial/bactericidal effects regardless of their association with additional antibacterial agents.

Furthermore, other research groups have also developed strategies for recycling antibiotics that are no longer functioning, taking advantage of NPs. This approach could allow researchers to avoid some procedures in terms of the research and regulatory issues, which are instead mandatory in the case of the development of new antibacterial molecules. In this regard, Si et al. recently proposed a new biodegradable and biocompatible chitosan-derived cationic antibacterial polymer with promising in vitro and in vivo antibacterial activity against MDR bacteria including *Listeria monocytogenes*, *S. aureus*, *E. coli*, *K. pneumoniae*, *P. aeruginosa* and *A. baumannii*. Notably, it showed a synergistic effect with many clinically relevant antibiotics, including amikacin, tobramycin, novobiocin, rifampicin and tazobactam [175]. In addition, Gupta et al. developed NPs able to block the efflux pumps of bacterial membranes, allowing them to regulate the dosage of antibiotics whose optimal efficacy depends on the choice of the cationic surface of the NPs [176]. 

Additionally, cationic NPs, and particularly dendrimers, with a cationic surface and a non-charged inner matrix, have been recently developed. These formulations possess both hydrophobic internal cavities, able to host hydrophobic small molecules, and hydrophilic exterior functions, which make them highly soluble in water. These macromolecules, in addition to interacting with the bacterial surface, can encapsulate water-insoluble bioactive molecules, otherwise not administrable and not clinically applicable. In this context, ursolic acid or pyrazole derivatives known to be endowed with several pharmacological activities have been successfully encapsulated in fourth- and fifth-generation cationic dendrimers modified with lysine, obtaining drug-loaded nanocomposites with proven high-water solubility and loading capacity [177,178,179]. When tested on bacteria, these nanocomposites displayed potent antibacterial activity, much higher than that of the encapsulated molecules, against Gram-positive MDR isolates of the *Staphylococcus* [180] and *Enterococcus* [181] genera, regardless of the intrinsic antibacterial effects of the dendrimers used as encapsulating agents. This achievement can be explained assuming that the cationic property of the NPs enveloping the bioactive molecules allowed their accumulation on the surfaces of bacteria and promoted their entry into the cells via pore formation. 

#### 6.3.2. Antitumor Function of Cationic NPs

The compounds and mechanisms of action of cationic NPs used to fight MDR cancer cells are very similar to those used against MDR bacteria. In fact, the use of NPs seems to be promising due to their ability to interact preferentially with the negatively charged membranes of cancer cells, thereby preserving the neutrally charged membranes of healthy cells (Figure 4). Indeed, cationic NPs for cancer therapy can be obtained through polyplexes, which are spontaneous complexes obtained either from the condensation of cationic polymers, directly from cationic polymers, by the coating of non-cationic NPs through copolymerization or by electrostatic interactions with cationic polymers, lipids, surfactants, or dendrimers [182]. 

Recently, Zhang et al. synthesized a novel compound consisting of cationic NPs based on hyaluronic acid encapsulated with docetaxel, which showed promising effects on MDR breast cancer cells. Specifically, this combination of substances was able to reduce the levels of CYP1B1, an enzyme over-expressed in breast cancer and responsible for resistance, and to improve the bioavailability of the antitumor agent [183].

The great efficacy of these compounds against resistant cancer cells is a promising starting point for the development of new therapies based on cationic NPs. In this context, we recently synthetized and tested on Etoposide-sensitive and Etoposide-resistant neuroblastoma cells two polystyrene-based (P5, P7) cationic nanosized copolymers, containing primary ammonium groups. Both copolymers were able to reduce cell viability and to increase ROS production in both cancer cell lines, demonstrating greater potency against Etoposide-resistant ones [184]. The same copolymers were assayed on more than sixty isolates of both Gram-positive and Gram-negative species and showed ultra-broad-spectrum bactericidal activity [185,186], thus confirming their common mechanism of action on bacteria and cancer cells. 

## 7. Conclusions and Final Remarks

Despite continued efforts to develop new antibiotic and anticancer therapies, the onset of resistance limits the effectiveness of such approaches, having significant repercussions for the health of the world population. For this reason, in recent years, research has increasingly focused on identifying alternative strategies. In this context, the use of MDPs is receiving particular attention and, among these, AMPs and ACPs, of natural or synthetic origin, seem to be promising. In fact, these systems, being able to specifically act on bacteria or cancer cells, limit the drug-induced secondary toxic effects. However, it is necessary to consider that the clinical development of both AMPs and ACPs shows several challenges and limitations, with their delivery identified as the most limiting problem. Among polymeric NPs, cationic ones are emerging and could have a relevant impact, both functioning as efficient delivery systems and exerting potent antibacterial and anticancer effects not only mediated by the cell membrane injury. 

Based on these findings and considering that therapy resistance is an alarming concern for global health, great efforts are needed to design and optimize cationic NP-based therapies and to progress them to clinical application.

## Figures and Tables

**Figure 1 ijms-23-06108-f001:**
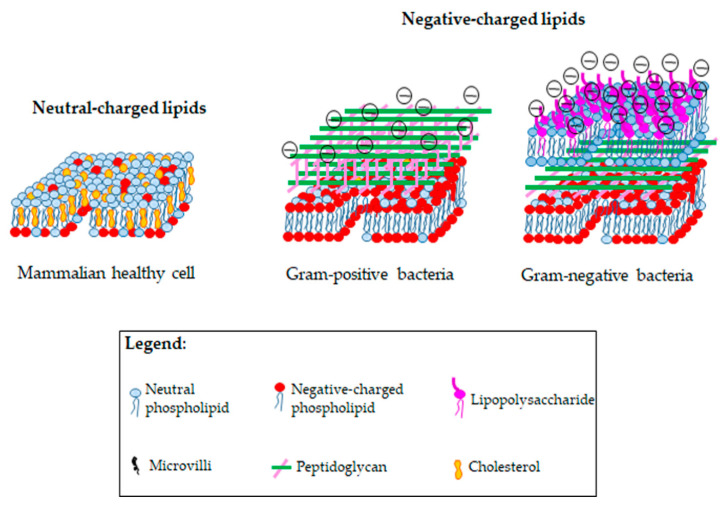
Comparison among membrane lipid composition of healthy mammalian cells, Gram-positive and Gram-negative bacteria.

**Figure 2 ijms-23-06108-f002:**
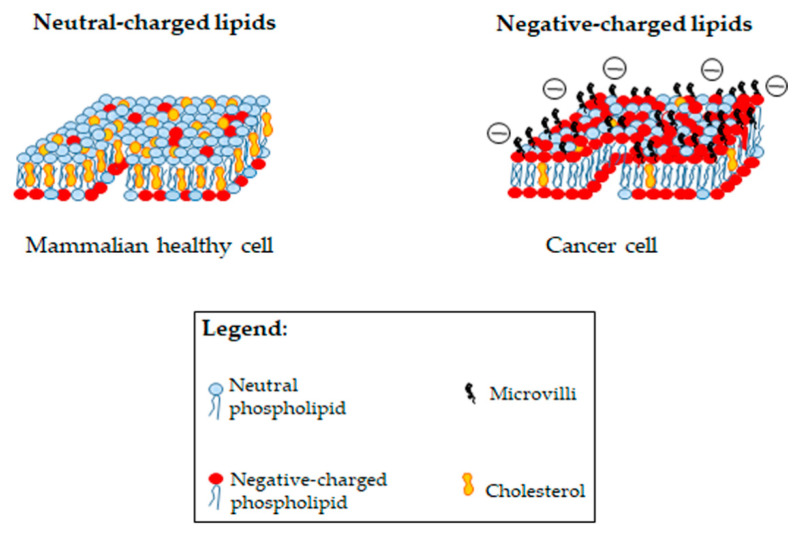
Comparison among membrane lipid composition of healthy mammalian cells and cancer cells.

**Figure 3 ijms-23-06108-f003:**
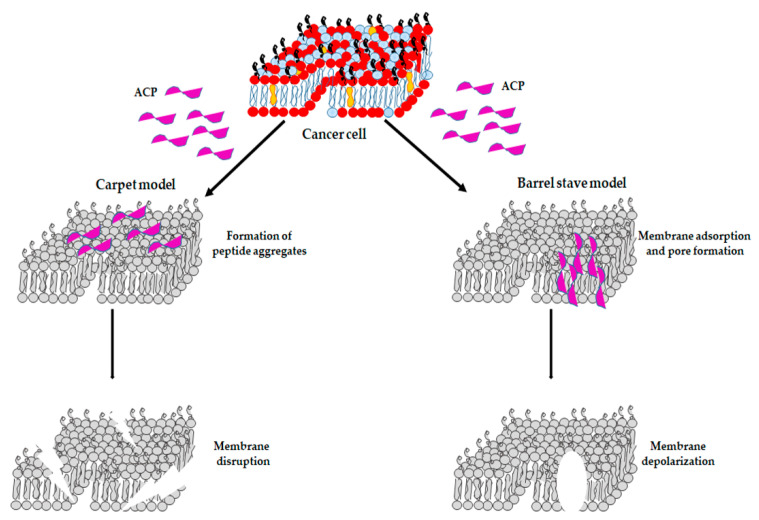
Models of ACPs’ action. In the carpet model (**left**), ACPs bind to the cell membrane via electrostatic interactions and, subsequently, ACPs may enter the cell membrane, inducing membrane disruption. In the barrel-stave model (**right**), peptides self-aggregate and form a transmembrane pore, leading to membrane depolarization.

**Figure 4 ijms-23-06108-f004:**
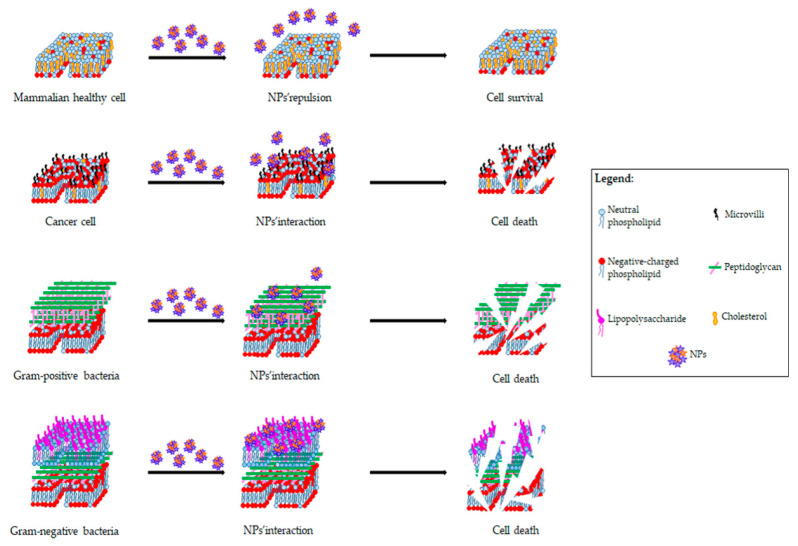
NPs’ interaction with cell membrane and their killing ability on cancer cells and bacteria.

**Table 1 ijms-23-06108-t001:** ACPs showing antitumor and/or antiviral effects, classified according to their structural characteristics into four categories: α-helical, β-pleated sheets, random coil and cyclic [116,117].

Category	Feature	ACP	Model
α-Helical [118,119,120,121]	Peptides short in length Simple in structure	Magainin II	Lung cancer cells (A549)
		Aurein	Glioblastoma cells (T98G)
		L-K6	Breast cancer cells (MCF-7)
		LL37	Colorectal cancer cells (HCT116)
		FK-16	
β-Pleated sheet [122,123,124,125,126]	Two or more disulfide bonds Good stability Structures more complex than α-helical	Bovine lactoferrin (LfcinB)	Gastric cancer cells (MGC803)
		MPLfcinB6	Human T leukemia cells
		MPLfcin-P13	Hepatocellular carcinoma cells (HepG2)
		Human neutrophil peptide (HNP-1)	Prostate cancer cells (PC-3)
Random coil ACPs [127,128,129,130,131]	Rich in proline and glycine Lack of a typical secondary structure	Alloferon	Herpes simplex virus Human papillomavirus
		KW-WK	N.R.
		PR39	N.R.
		PR35	N.R.
Cyclic ACPs [132,133,134]	Closed peptides composed of a head-to-tail cyclization backbone or disulfide bonds that form cystine knots More stable than linear structures	Hedyotis diffusa Cytide 1–3	Prostate cancer cells (PC3, DU145, LNCap)
		H-10	Malignant melanoma cells (B16)
		RA-XII	Colorectal tumor cells (HCT116)

N.R. = not reported.

**Table 2 ijms-23-06108-t002:** ACPs in clinical trials ongoing or already approved and marketed.

ACPs in Development	Phase of Development	Target Tumor	Advantages
Bryostatin 1 [135,136]	Phase 1	Melanoma Lymphoma Ovarian carcinoma	
Aplidine (plitidepsin) [137]	Phase 1 (completed)		Well tolerated Low toxicity
	Phase 2 (in progress)	Advanced medullary thyroid carcinoma Advanced malignant melanoma Small-cell lung cancer Advanced renal carcinoma	
Kyprolis [115]	Approved by FDA and EMA (marketed)	N.R.	N.S.
SomaKit TOC [115]			
Lutathera [115]			
Gallium Dotatoc Ga68 [115]			

N.R. = not reported; N.S. = not specified.

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
