# Peer review of "Antimicrobial Peptides and Cationic Nanoparticles: A Broad-Spectrum Weapon to Fight Multi-Drug Resistance Not Only in Bacteria"

_ijms, 2022, doi:10.3390/ijms23116108_

Round 1
Reviewer 1 Report
The review article seems to nicely summarize the research of antimicrobial peptides and catanionic nanoparticles with regards to multi-drug resistance in bacteria and cancer cells. I think the authors should address the below points before publishing this review article in the IJMS journal.
1. At lines 105-108: Can the specificity of antimicrobial peptides be explained merely by the difference of lipid compositions? I understand that the outer leaflet of mammalian cell membranes is mainly composed of lipids with zero net charges. However, at the cell surface express proteoglycans that abundant of negatively charged glycosaminoglycans, such as heparan sulfate.
2. At lines 114-115: Is the statement “the primary target of NAMPs is the external envelope of bacteria” well established? Should cite some references with this description.
3. At lines 129-131: Could you add some explanation for the elevated hemolytic toxicity of AMPs? Is this derived from their cationic charge that leads to binding to the cell membrane because of the presence of negatively charged glycosaminoglycans at the surface?
4. At lines 142-143 and 146-148: The mammalian cells including healthy ones, have a negatively charged cell surface because of sulfated glycosaminoglycans ubiquitously expressed at the cell surface. Therefore, it would be difficult to state the sentences at lines 146-148 unless there are some references/studies to support this idea that AMPs preferably bind to cancer cells compared with healthy cells. I thought that non-specific binding of AMPs to cells cause the issue of their clinical development, which leads to the idea that utilizes the AMP delivery with nanoparticles as discussed in the later part of this review.
Author Response
REVIEWER 1
The review article seems to nicely summarize the research of antimicrobial peptides and catanionic nanoparticles with regards to multi-drug resistance in bacteria and cancer cells. I think the authors should address the below points before publishing this review article in the IJMS journal.
We thank the Reviewer for the appreciation and the useful comments that we have taken into consideration in the new version of the review article.
- At lines 105-108: Can the specificity of antimicrobial peptides be explained merely by the difference of lipid compositions? I understand that the outer leaflet of mammalian cell membranes is mainly composed of lipids with zero net charges. However, at the cell surface express proteoglycans that abundant of negatively charged glycosaminoglycans, such as heparan sulfate.
We thank the Reviewer for the comment and we try to satisfy the request. As we have explained in the revised version of the manuscript, the specificity of AMPs is mainly due to the different membrane lipid composition of bacterial and eukaryotic cells (lines 105-112). In fact, the electrostatic interactions between NAMPs and the negative-charged lipids of the bacterial surface such as phosphatidylglycerol (see Ref 46) are the main responsible for the higher selectivity of NAMPs towards bacteria. The lower specificity of NAMPs towards mammalian cells is due to the fact that the outer leaflet of their membrane is principally composed of lipids with zero net charge, such as phosphatidylcoline (see Ref 46). With regard to the statement that specificity of antimicrobial peptides is due to the difference of membrane lipid composition we have reported 10 references (see Refs 38-47) related to this issue.
- At lines 114-115: Is the statement “the primary target of NAMPs is the external envelope of bacteria” well established? Should cite some references with this description.
As above explained, the electrostatic interactions between NAMPs and the negative-charged lipids of the bacterial surface are the main responsible for the higher selectivity of NAMPs towards bacteria (see Refs 38-47) and the sentence was modified as follows: “…..… the primary target of NAMPs is the lipid matrix of bacterial cell membranes [46]
- At lines 129-131: Could you add some explanation for the elevated hemolytic toxicity of AMPs? Is this derived from their cationic charge that leads to binding to the cell membrane because of the presence of negatively charged glycosaminoglycans at the surface?
With regard to the sentence reported at lines 129-131 of the original manuscript, the hemolytic activity of AMPs is correlated to peptide hydrophobicity (see ref. 62). In fact, polar peptides, although endowed with significant cytotoxic effects have been found to have a low hemolytic activity. Moreover, the ability of some AMPs to form amphipathic solution structures is linked to an enhanced antimicrobial activity but also to an increased hemolytic activity once a threshold hydrophobicity has been reached (see Refs 63, 64). As explained before and according to the references cited in the text, AMPs specifically interact with the negative-charged lipids of the cell membrane and the interaction with negative-charged glycosaminoglicans is not reported in the literature. This could be an interesting point of future studies.
- At lines 142-143 and 146-148: The mammalian cells including healthy ones, have a negatively charged cell surface because of sulfated glycosaminoglycans ubiquitously expressed at the cell surface. Therefore, it would be difficult to state the sentences at lines 146-148 unless there are some references/studies to support this idea that AMPs preferably bind to cancer cells compared with healthy cells. I thought that non-specific binding of AMPs to cells cause the issue of their clinical development, which leads to the idea that utilizes the AMP delivery with nanoparticles as discussed in the later part of this review.
We thank the Reviewer for this comment and we try to better explain this point. The specificity of AMP’s binding with bacterial cell membrane and cancer cell membrane is due to the major presence in both membranes of negative-charged lipids (see Refs 75, 84-86) in comparison with the predominance of neutral-charged lipids in the membrane of healthy mammalian cell. In order to avoid misunderstanding, we have modified the sentence that is reported at lines 152-155 of the revised version. With regard to the issue of NAPM’s and SAMP’s clinical development rightly outlined by the Reviewer, we have discussed in the sections 5 and 6 explaining the reasons that led to the design of nanoparticle-based delivery systems.
Reviewer 2 Report
The review reports on novel strategies to defeat multi-drug resistance both in bacteria and cancer cells based on the employment of nanoparticles. The review is well written, and the topic is relevant.
Some comments that need to be addressed by authors are reported below:
- More recent articles on the topic should be reported by authors, such as: 10.1038/s41579-022-00700-5; 10.1038/s41570-022-00373-x; 10.1038/s41568-022-00454-5.
- I suggest adding the term “antimicrobial peptides” to the keywords.
- One the most recent approaches against multidrug resistance is based on the development of nanozymes that are obtained by conjugating active compounds or proteins with nanomaterials, see for example: 10.3389/fmicb.2018.01441; 10.1002/adfm.201900518; 10.1016/j.jcis.2020.07.006; 10.1007/s12274-020-2824-7.
This aspect should be discussed by authors to enrich the list of possible strategies reported in the review. - Concerning the factors that could influence cancer cell response with respect to normal healthy cells, also the overexpression of folic acid receptors on the cancer cell membrane can be exploited for targeted drug delivery, see for instance: 10.4161/cbt.22020; 10.1073/pnas.1308827110; 10.1039/c9nr01075k.
- The caption of Table 1 reports: “ACPs showing anti-tumor and/or antiviral effects”, but the antiviral action of the ACPs is not discussed in the text.
- I suggest adding a sketch representing the two main mode of action of ACPs, “barrel-stave” and “carpet” (line 214-223) to help the readers to visualize the interaction mechanisms of ACPs with cell membrane.
Author Response
REVIEWER 2
The review reports on novel strategies to defeat multi-drug resistance both in bacteria and cancer cells based on the employment of nanoparticles. The review is well written, and the topic is relevant.
We thank the Reviewer for the appreciation and the useful comments that we have taken into consideration in the new version of the review article.
Some comments that need to be addressed by authors are reported below:
- More recent articles on the topic should be reported by authors, such as: 10.1038/s41579-022-00700-5; 10.1038/s41570-022-00373-x; 10.1038/s41568-022-00454-5.
Although we believe that it is not correct to require the inclusion of specific references, in order to satisfy the Reviewer's requests, we have decided to include the paper by Chagri et al (10.1038/s41570-022-00373-x) because we have considered it pertinent for the herein study.
- I suggest adding the term “antimicrobial peptides” to the keywords.
Taking into account the Reviewer’s suggestion, “antimicrobial peptides” was added to the keywords.
- One the most recent approaches against multidrug resistance is based on the development of nanozymes that are obtained by conjugating active compounds or proteins with nanomaterials, see for example: 10.3389/fmicb.2018.01441; 10.1002/adfm.201900518; 10.1016/j.jcis.2020.07.006; 10.1007/s12274-020-2824-7. This aspect should be discussed by authors to enrich the list of possible strategies reported in the review.
As explained at the point 1, we have decided to include the manuscript by Baptista et al. (10.3389/fmicb.2018.01441) since it gives an interesting overview about the use of nano-strategies to fight antibiotic resistance.
- Concerning the factors that could influence cancer cell response with respect to normal healthy cells, also the overexpression of folic acid receptors on the cancer cell membrane can be exploited for targeted drug delivery, see for instance: 10.4161/cbt.22020; 10.1073/pnas.1308827110; 10.1039/c9nr01075k.
We thank the Reviewer for the suggestions, but we have decided to not include the proposed references because the article review, in agreement with the aim of this Special Issue, is focused on cationic nanoparticles. In addition, if we consider folic acid receptors as targets useful for specific drug delivery, we should also include several other molecules mainly expressed on cancer cell surface, but this is not the focus of this paper.
- The caption of Table 1 reports: “ACPs showing anti-tumor and/or antiviral effects”, but the antiviral action of the ACPs is not discussed in the text.
We apologize but we have reported the anti-viral action of ACPs in the Table 1 in order to show their broad spectrum of action.
- I suggest adding a sketch representing the two main mode of action of ACPs, “barrel-stave” and “carpet” (line 214-223) to help the readers to visualize the interaction mechanisms of ACPs with cell membrane.
We agree with the Reviewer’s suggestion and a sketch representing the two main mode of action of ACPs has been added (Figure 3 of the revised manuscript).